# Artificial Intelligence in Healthcare: University Students’ Perceptions and Level of Confidence

**DOI:** 10.3390/healthcare13182312

**Published:** 2025-09-16

**Authors:** Paulo Simões Peres, Luísa Castro, Ivone Duarte

**Affiliations:** 1Faculty of Medicine, University of Porto, 4200-319 Porto, Portugal; 2RISE-Health, Department of Population Studies, School of Medicine and Biomedical Sciences, University of Porto, 4200-319 Porto, Portugal; luisacastro@icbas.up.pt; 3Center of Bioethics of the Faculty of Medicine, University of Porto, 4200-319 Porto, Portugal; iduarte@med.up.pt; 4RISE-Health, Department of Community Medicine, Information and Health Decision Sciences (MEDCIDS), Faculty of Medicine, University of Porto, 4200-319 Porto, Portugal

**Keywords:** Artificial Intelligence (AI), trust, liability, legal, literacy, medical education, students

## Abstract

**Introduction/Objectives**: The continuous progress of information technologies and their increasing use in the health sector have driven the integration of these technologies into the care of the population, including the progressive use of Artificial Intelligence (AI). Given the rapid growth of AI, legislation and scientific evidence have been accompanying developments, clarifying the place of this technology in society. This study aimed to determine university students’ perspectives on the use of AI in healthcare, correlating them with sociodemographic characteristics. **Methods**: Data were collected using an original personal questionnaire to first-year students from four organic units at the University of Porto, between December 2024 and March 2025. **Results**: A total of 235 responses were obtained from four different Faculties, and no significant differences were found between gender, area of study, or course, regarding perspectives on the inclusion of AI in healthcare. Across the board, students view this inclusion positively, even though they trust a doctor more and do not have uniform positions regarding the system’s accountability. **Conclusions**: Thus, the study’s results highlight the need to deepen the debate and training on AI in healthcare, to promote the conscious, critical, and ethical integration of these technologies into healthcare.

## 1. Introduction

The advent of technologies in healthcare has changed the paradigm in recent decades, given their applicability in multiple scenarios [1]. According to the World Health Organization, health technologies are associated with the development and use of digital technologies to improve health [2], which can be correlated with greater precision and efficiency in the provision of healthcare, and a lower predisposition to error [3].

The emergence of Artificial Intelligence (AI) dates to the 20th century and can be defined as a semi-autonomous system that enables the computerized imitation of human cognitive functions [4,5]. Recent years have therefore been marked by a continuous evolution of AI technologies in the various aspects of healthcare provision—prevention, diagnosis, clinical decision-making, and treatment [6]. The advantages associated with its use have been widely described in the literature, due to the large amount of data through which healthcare algorithms can be developed, as well as the possibility of, among other capabilities, applying this data to real-time inferences, making it possible to estimate health risk and the best diagnostic and therapeutic hypotheses [5,7].

Nevertheless, widespread AI implementation involves numerous barriers, including disruptive technologies that arise in a well-established industry, such as healthcare [1]. These barriers are based on multiple aspects, such as informed consent, privacy, equity, access [1], and the lack of adequate regulatory mechanisms to ensure the quality and reliability of the implemented technologies [1,8]. In this context, the European Union approved the “AI Act” in 2024, the first legal framework to address the problems of this technology, making a risk-based analysis and seeking to create harmonized rules on Artificial Intelligence. In this sense, health is a high-risk use case, requiring strict measures to prevent harm [9].

Due to the multiplicity of factors involved, the inclusion of AI in healthcare provision is perceived differently by various societal groups, including patients, medical students, and doctors [10,11,12]. About patients, Fritsch, S.J. et al. suggest that trust in healthcare professionals predominates, to the detriment of AI [10]. As for doctors, the main barriers identified concern information communication to patients, the ability to use AI tools, and patient safety [13]. Specifically among Portuguese doctors, the influence of age on the perception of the impact of AI stands out, as does the importance of training for its use during undergraduate medical education [14]. Meanwhile, medical students recognize the importance of AI and its usefulness in complementing the functions performed by clinicians [12]. The existing body of evidence underscores the need to incorporate AI into undergraduate healthcare training, while also raising awareness of the potential ethical issues associated with its use [15,16].

Moreover, multiple sociodemographic factors seem to be implicated in the use of AI: female sex and a low level of education are associated, on the one hand, with a lower technical affinity for these technologies and, on the other, with a more negative perception of them [10]. The impact of prior training in the use of AI is also noteworthy, showing it to be a predisposing factor for greater confidence and positivity in the perception of AI [17]. Furthermore, Pedro, A.R. et al. consider that the cultural and contextual specificities of the populations under study cannot be overlooked regarding perceptions of AI [17].

The controversy associated with multiple topics related to AI and the lack of high-level evidence denote the need for more studies on AI and the various factors influencing perceptions of its inclusion in healthcare [18]. Despite the growing body of literature on perceptions of AI in healthcare, few studies have focused on specific academic communities, particularly within the Portuguese context. Existing research often generalizes perceptions without accounting for the cultural, educational, and institutional specificities that may significantly shape attitudes toward AI.

This study aims to ascertain the perceptions and confidence of the student community of a Portuguese university regarding the inclusion of Artificial Intelligence tools in healthcare, as well as to determine which factors are related to the diversity of these attitudes.

## 2. Methods

The present study is a cross-sectional observational study, with data collected through the in-person use of a paper questionnaire.

### 2.1. Questionnaire

The questionnaire was developed for this study, given the lack of validated questionnaires in Portuguese with the same objectives. It was divided into a section on sociodemographic data, general questions about AI, and AI in healthcare (Appendix A).

Regarding sociodemographic data, questions were included that had been previously assessed in studies of the same population and which proved to be relevant [19], such as age (question 1), gender (question 2) and area of study (questions 3 and 4), as well as additional questions to allow an assessment of factors influencing the students’ perspectives, namely the country in which they attended high school (question 5) and whether they attended public or private education during the same period (question 6).

To quantify students’ use and knowledge of AI as a whole and their views on the literacy of the Portuguese population, general questions were included about the definition of AI (questions 7 and 8), students’ average use (question 9), and society’s digital literacy (questions 11 and 12).

Existing evidence was collected for the creation of the questionnaire, based on studies already carried out, as well as original questionnaires, with patients [10], health professionals [17], and students [20]. The questionnaires already validated in AI were also revised to include questions about assessing trust and perspectives on including this technology in care [21,22]. A Likert scale was used on the applicable questions with the following response options: 1: Strongly disagree; 2: Disagree; 3: Neutral; 4: Agree; 5: Strongly agree.

### 2.2. Questionnaire Validation

Before administering the questionnaire and collecting responses, a pre-test was carried out at the Faculty of Medicine of the University of Porto with 27 s-year medical students in November 2024, representing a population similar to the target of the study. It included questions about the relevance and clarity of each question, time taken to complete the questionnaire, and a field for written suggestions or comments. Sampling was carried out by convenience, through presence in the classroom; the same process was used to collect responses afterwards. The Content Validity Index (CVI) was calculated to assess clarity and relevance. Participants were asked to score each question from 1 to 4, depending on how clear (1: Not clear; 2: Item needs some revision; 3: Clear, but needs minor revision; 4: Very clear) and relevant (1: Not applicable; 2: Item needs some revision; 3: Relevant, but needs minor revision; 4: Very relevant) they perceived it to be. The CVI was calculated for each question (I-CVI) through the ratio between the sum of the number of “3” and “4” answers and the number of valid answers, and for the scale (S-CVI) through the average of the I-CVI, with the data shown in Table 1 [23].

Regarding the time taken to fill in the form, 20 responses were obtained because the remaining participants did not fill in the blanks. The average time taken to fill in the form was 7 min and 45 s, with the minimum time reported being 4 min and the maximum 11.

Regarding the open-ended question, only one was submitted, referring to the potential lack of knowledge of the concept of “medical confidentiality” (question 31) by the general population. Nevertheless, no changes were made to the wording of the item, as it was considered a term widely disseminated by the community, particularly among higher education students. Furthermore, the question was deemed clear by the I-CVI.

### 2.3. Target Population and Sample

The target population for this study consists of first-year university students from different Organic Units at the University of Porto. This population was selected to obtain a general overview of young people’s perspectives on AI in healthcare. By selecting first-year students, the aim was to reduce any bias that higher education in different areas might cause in each student’s individual view and avoid introducing new information through university education. In addition, it sought to reach students from various regions of knowledge to understand whether the previous interests that led them to choose a particular course in higher education were related to other perspectives on AI. It was therefore initially planned to collect data from six Organic Units (OUs): the Faculty of Medicine, the Nursing School, the Faculty of Psychology and Education Sciences, the Faculty of Law, the Faculty of Engineering, and the Faculty of Architecture. This choice would make it possible to understand the potential differences between students in health fields and students in other areas, particularly Law and engineering, domains with a relevant relationship to the advent and challenges of AI.

To collect the data in the institutions, after approval by the Ethics Committee of the Faculty of Medicine of the University of Porto, the Deans of each OU were contacted to request authorization to carry out the study. As confirmation was not received from all the above, or was not received promptly, the Faculty of Dental Medicine and the Porto School of Engineering were also contacted. Even so, it was impossible to collect data from all the planned OUs, being carried out at the Faculty of Medicine, the Faculty of Architecture, the Faculty of Psychology and Education Sciences, and the Faculty of Law, guaranteeing a sample of students from health and other areas.

Sampling was conducted conveniently, selecting students based on their attendance at previously defined classes, with the authorization of the Deans of the OUs and the respective professors. No power calculation was performed before determining the sample size, as this pilot study intended to provide preliminary insights into students’ perceptions of AI in healthcare and inform future research with larger and more representative samples. However, based on the sample size (*n* = 235), the maximum margin of error for proportion estimates, assuming a reference proportion of 0.5 and a 95% confidence level, was approximately 6.4%. The fact that recruitment took place in-person, in the classroom, allowed for greater randomization in the selection of participants, reducing the risk of bias arising from the exclusive participation of individuals with prior familiarity or interest in the subject. It should be noted that at the Faculty of Psychology and Education Sciences of the University of Porto, the class in which the questionnaire was administered was compulsory, unlike the other faculties involved (FMUP, FDUP, and FAUP), where the class was optional.

Participation was voluntary, preceded by clear information about the study’s objectives, guaranteeing the anonymity and confidentiality of the data collected. The questionnaire was administered in physical format, between December 2024 and March 2025.

### 2.4. Inclusion and Exclusion Criteria

All adult (at least 18 years old) university students attending their first year at the selected units (FMUP, FAUP, FPCEUP, and FDUP) were considered eligible for inclusion in the study, regardless of the course they were taking, as long as they belonged to the same OU. All participants provided written informed consent prior to completing the questionnaire and retained a copy of the document.

### 2.5. Bias Reduction

To reduce potential bias, the data were collected in a classroom context, ensuring random sampling and avoiding the selection of participants with a prior interest in the topic. In addition, the selection of first-year students was an attempt to minimize the influence that higher education and the knowledge acquired during it could have on the opinions of those surveyed.

### 2.6. Statistical Analysis

The data collected on paper were entered into a computerized database, initially in Microsoft^®^ Excel (Microsoft, Redmond, WA, USA) and later exported for statistical analysis in IBM SPSS Statistics^®^ version 29.0 (IBM Corp., Armonk, NY, USA) software. The statistical analysis included descriptive measures to characterize the sample: absolute (*n*) and relative (%) frequencies for categorical variables; median with interquartile range for quantitative variables with a distribution deviating from normal. Statistical tests were carried out to compare independent groups according to the nature of the variables: Chi-square test or Fisher-Halton exact test (when the count was less than 5 in more than 20% of the expected values) for categorical variables, Mann–Whitney (2 groups) and Kruskal–Wallis (3 or more groups) tests for independent variables. Bonferroni correction was applied for multiple comparisons after a Kruskal–Wallis test with a significant test value. In all the statistical tests, a *p*-value below 0.05 was interpreted as evidence of a statistically significant association or difference, suggesting that the observed results are unlikely to be due to chance alone

### 2.7. Ethical Considerations

The study was conducted per the ethical principles of the Declaration of Helsinki and complied with the General Data Protection Regulation (GDPR). All participants were duly informed about the purpose of the study and signed an informed consent form before participation, which could be withdrawn at any time by contacting the researcher. Approval to carry out the study was obtained from the Ethics Committee of the Faculty of Medicine of the University of Porto (ref. 294/CEFMUP/2024).

## 3. Results

The sample consisted of 235 first-year university students from four OUs at the University of Porto: Faculty of Medicine (*n* = 67; 28.5%), Faculty of Psychology and Education Sciences (*n* = 19; 8.1%), Faculty of Law (*n* = 102; 43.4%), and Faculty of Architecture (*n* = 47; 20%). All participants in the chosen classes agreed to participate in the study and signed the informed consent form, resulting in a 100% response rate.

This resulted in 235 valid responses, whose sociodemographic characteristics are shown in Table 2.

### 3.1. Sociodemographic Characteristics (Q1–Q6)

Most of the participants were female (74.9%), completed their secondary education in Portugal (85.9%), and mainly in public schools (65.8%). The sex distributions showed no statistically significant differences (χ2 (3) = 2.806; *p* = 0.423), as did the variables related to the type of school (χ2 (3) = 0.946; *p* = 0.814) and country of secondary education (χ2 (3) = 2.611; *p* = 0.456), when compared between the Faculties. Table 2 shows the sociodemographic characterization of the sample by Faculty.

### 3.2. Knowledge, Use, and Familiarity with Artificial Intelligence (Q7–Q12)

Almost all the students (96.2%) correctly identified the definition of Artificial Intelligence (question 7), and the comparison between faculties in terms of the proportion of correct answers revealed no statistically significant differences (*p* = 0.193). Regarding the participants’ self-reported knowledge (question 8), at least 75% reported Basic Knowledge (3) or lower.

Regarding the participants’ use of AI (question 9), 34.1% of the participants reported frequent (4) or very frequent (5) use of this type of tool. When a comparative analysis was carried out between the participants’ OUs, significant differences were found (H(3) = 9.722; *p* = 0.021), which refer to differences in perception of the use of AI between FAUP and FMUP (*p* = 0.024), with the latter’s students using it more frequently.

Regarding the self-assessment of familiarity with AI in health (question 10), the results show that at least half of the students disagreed (2) or strongly disagreed (1) with the statement “I am familiar with the application of AI in health.”. No significant differences were found between sexes (U = 5115; *p* = 0.857) or between students from the health area (FMUP and FPCEUP) and the other regions (FDUP and FAUP) (U = 5575; *p* = 0.080).

As for the perception of Portuguese society’s preparedness to integrate AI, most participants gave neutral or disagreeing answers. For question 11, “The Portuguese society is ready to implement AI tools.”, at least 25% of participants said they disagreed (2) or strongly disagreed (1) with the statement. This figure rises to at least 50% of the participants in the disagreement (2) or intense dispute (1) responses regarding the perception of the Portuguese population’s digital literacy (question 12). It should be noted that, in the case of question 11, zero participants totally agreed with the statement, and only one participant in the second question, corresponding to 0.4%.

### 3.3. Expectations Regarding AI in Healthcare (Q18, 21, 22, 23, 33, and 34)

Regarding specific views on the inclusion of AI in healthcare, the following questions were analyzed: “AI will improve healthcare.”, “I trust AI more than a doctor’s opinion.”, “AI will reduce medical errors.”, “AI will enable more accurate diagnoses.”, “The doctor using AI is responsible for any errors arising from its use,” and “AI programmers are responsible for any errors arising from its use.” The results are shown in Figure 1 and Table 3.

The answers to these same questions were also compared between sex and between health courses and other areas, and no significant differences were found, as shown in Table 3.

Regarding the advantages of using AI in healthcare, 57.4% agree (4 or 5) with the statement about AI improving healthcare (question 18). However, regarding reducing medical errors, only 37% agreed (4 or 5) with the statement (question 22). No significant differences were found between the sexes or health and non-health students.

Concerning the responsibility for errors arising from using AI, at least half of the participants agreed (4) or totally agreed (5) that the doctor was responsible. On this question, at least 25% of males and non-health students agree (5) that the responsibility lies with the doctor, while this figure is lower among females and health students. Even so, no significant differences were found between groups in these variables (U = 4766.5 and *p* = 0.328 for sex; U = 6397 and *p* = 0.985 for area of study).

In the case of attributing responsibility to programmers, at least half of the participants reported a neutral (3) or disagreeing (1 and 2) position, with no statistically significant differences found (U = 5078 and *p* = 0.952 for sex; U = 5476 and *p* = 0.061 for area of study).

In summary, students showed high general knowledge regarding AI, but low familiarity with this tool in healthcare. They expressed trust in doctors over AI, while views on accountability for AI-related errors were divided. Only one significant difference emerged between faculties (FMUP vs. FAUP) in AI use, and none between sexes.

## 4. Discussion

This study sought to understand university students’ perceptions and attitudes towards using AI in healthcare. It also explored their relationship with the students’ sociodemographic characteristics, such as their field of study and sex. Unlike other existing studies, various fields were included—inside and outside healthcare—further deepening the evidence regarding the general perception of AI in healthcare.

The data analysis revealed several relevant trends, albeit without statistically significant differences between groups, which might provide an essential insight into youth literacy and perceptions of AI in Portugal, specifically at the University of Porto.

### 4.1. Literacy and Knowledge

Most students correctly identified the definition of AI, which aligns with the results of other studies that point to high generic knowledge among young and educated audiences [24]. However, when asked to self-assess their familiarity with AI applications in health, a majority disagreed with a statement regarding their familiarity with the topic. This alignment suggests that the perception found in Portugal follows an international trend observed among university populations, where theoretical familiarity with AI tends to be widespread, even if concrete experience with applied tools in healthcare remains limited. This dissociation between conceptual and operational knowledge has been documented in previous literature, including medical contexts [12,25]. This indicates that curricula should move beyond generic digital literacy, including applied AI training for healthcare students and future professionals.

The inexistence of a greater familiarity in health students—although an apparent trend is present—may denote an absence of specific education on AI for healthcare students, even when focusing on the implications for their future profession. Even so, given the topic’s relevance, promoting particular training on digital tools is essential. In fact, some authors point out the absence of this literacy regarding the specific use of AI in healthcare, although present in a generic way for AI [26].

### 4.2. Trust and Perception on the Impact of AI

Regarding expectations, the results reflect a predominantly positive perception, already documented in other studies with university students. At the same time, the persistence of greater trust in human clinical judgment, coinciding with international studies, suggests that these attitudes are not merely local but instead part of a broader international pattern in which the irreplaceability of the medical profession remains a central belief [12].

On the other hand, the perception regarding the literacy of the Portuguese society is congruent with the national reality, given that, according to the European Commission’s Digital Economy and Society Index, Portugal remains at the European average in basic digital skills, with room for improvement [27].

### 4.3. Liability and Ethical Concerns

Regarding the liability when using AI systems, although not consensual, the participants tend to hold the doctor liable to the detriment of the programmers. This perception reflects the ethical ambiguity in the debate on medical–technological responsibility [28]. On the one hand, this responsibility depends on the tool being used, and it is essential to analyze whether it is autonomous or a decision support system. In the latter case, the legal framework considers that responsibility falls on the professional who makes the final decision.

On the other hand, it is also essential to think that, since it is a created product, it could be defective and, in this case, the responsibility would fall on its programmers [29]. Across the board, it is also essential to consider the concept of the “black box” when considering the legal responsibility of using AI. This concept reflects the opacity of AI systems, where it is often impossible to understand the “thinking” mechanism of the process being carried out by the system. Although this can bring advantages, such as the system replicating biological phenomena that are not yet fully understood, it prevents a clear view of how the process works, making it challenging to create legal mechanisms to determine the origin of medical error, for example, [30].

Considering this, it is essential to highlight the developments brought by the European AI Act, which sets requirements for high-risk uses of AI, such as those that impact health. Among others, it mandates the system to be accurately overseen by humans during its use. This considers the importance of ensuring the user can interpret the system’s output, override it, and be aware of the possibility of automation bias—overreliance on the system without proper critical interpretation [9]. Therefore, developing laws and rules on transparency and human control over AI may fill the existing gap in regulation and ultimately ensure that medical professionals assume responsibility for their use of AI technologies while maintaining a cautious approach [31]. These tools may only be widely adopted in healthcare once an appropriate legal framework and safeguards are in place. Nonetheless, given the current level of students’ knowledge, it is essential that such topics be incorporated into higher education curricula.

### 4.4. Lack of Statistical Differences and Possible Interpretations

The sample size and data collection constraints may have limited the detection of actual effects across subgroups. Given the small number of participants in some OUs, future studies should aim for more comprehensive sampling with larger and more representative cohorts. This may require establishing collaborations directly with the OUs or the University and extending the data collection period.

Although it was possible to obtain information about the students’ perspectives on AI in healthcare, few tests revealed statistical significance when comparing the sociodemographic characteristics chosen. In fact, only the comparison of AI use between FMUP and FAUP showed significant differences, and the greater use at FMUP could be explained by greater exposure to digital tools applied to health and the sciences. However, the other comparisons (by sex, course, and study area) revealed no significant differences.

The absence of statistically significant differences should be interpreted with caution. On the one hand, they may reflect the homogeneity of perceptions in a student population that has already achieved higher education and therefore has a relatively high level of education. On the other hand, this homogeneity could also be explained by the increasing exposure to technological discourse among younger people and the population.

### 4.5. Limitations

One of the main limitations of this study is the sample size. In fact, the barriers encountered in obtaining authorization from the OUs, as well as the dependence on the presence of students in the classes where the questionnaires were administered—due to convenience sampling—resulted in a reduction in the sample size and may limit the generalization of the data obtained to the target population. In addition, the distribution of participants was unbalanced in terms of gender, with a predominance of female students. Despite using non-parametric statistical tests to minimize this bias, this disproportion may have influenced some of the perceptions analyzed, despite reflecting the predominance of female students in the OUs involved. Moreover, the optionality of the attendance of the classes may create a potential bias, given that attending courses of this nature is associated with greater commitment to the academic field. In future studies, it may be relevant to determine whether there is a relationship between academic performance and perspectives on AI.

The study’s design was also a limitation, as it does not allow causal relationships to be established or ensure the representativeness of the Portuguese higher education student population. In the future, different study designs with larger samples must be performed to ensure the data reflect the students’ perceptions, ensuring greater balance between OUs, field of study, and sex.

Although the sample aimed to select students from the health area and other areas, it fell short of what was expected. It would be relevant in the future to diversify the sample, namely by including students from more areas of knowledge—inside and outside health—as well as students from the areas of technology, engineering, and data science. Also in this context, it may be relevant to include students from different OUs, even though they are studying the same courses, reducing the potential confounding factor of the place of study.

## 5. Conclusions

This study aimed to understand university students’ perceptions regarding integrating Artificial Intelligence into healthcare. Despite their familiarity with AI tools, there was little specific experience and literacy in the health field, highlighting the importance of including targeted training on these technologies in academic curricula in the health field, particularly in medicine. Perceptions were generally positive, with reservations about replacing human clinical judgment and uncertainties about liability for errors or failures in using this technology.

The lack of statistically significant differences between groups may reflect a homogenization of perspectives in a young generation that is transversally exposed to technological discourse and tends to have high digital literacy. However, limitations such as the size and composition of the sample should be considered when interpreting the results and considered in future studies.

Beyond enhancing our understanding of students’ perceptions, this study offers valuable insights for universities and policymakers in shaping educational and regulatory strategies. The integration of artificial intelligence in healthcare should be gradual and responsible, balancing innovation with caution. In this context, fostering ongoing dialog between academia, healthcare professionals, and legislators is essential to ensure that the future of AI-supported clinical practice is not only technologically advanced but also ethically sound and patient-centered.

Future studies, with larger and more diverse samples, could delve deeper into these issues and contribute to developing appropriate educational and regulatory tools and the careful progressive integration of AI into the population’s healthcare.

## Figures and Tables

**Figure 1 healthcare-13-02312-f001:**
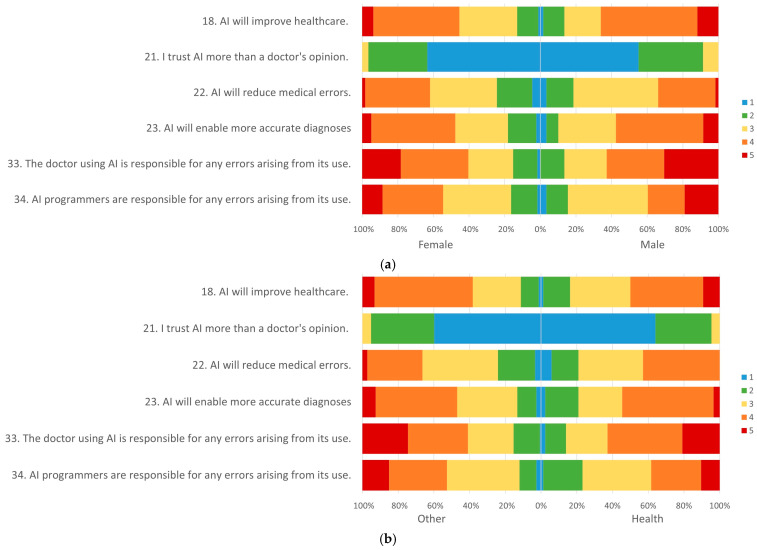
Frequency of the answers regarding expectations regarding AI in healthcare: (**a**) By sex; (**b**) by area of study. (1: Strongly disagree; 2: Disagree; 3: Neutral; 4: Agree; 5: Strongly agree).

**Table 1 healthcare-13-02312-t001:** Clarity and relevance of the 28 questions, based on the 27 participants in the pre-test.

	Clarity		Relevance	
Question	I-CVI	S-CVI	I-CVI	S-CVI
7	1.00		1.00	
8	1.00		1.00	
9	1.00		1.00	
10	1.00		0.96	
11	1.00		0.92	
12	1.00		0.92	
13	0.96		0.96	
14	0.96		1.00	
15	1.00		0.96	
16	1.00		0.96	
17	1.00		1.00	
18	1.00		1.00	
19	1.00		1.00	
20	1.00		1.00	
21	1.00		1.00	
22	1.00		1.00	
23	1.00		1.00	
24	1.00		1.00	
25	0.96		1.00	
26	1.00		0.96	
27	1.00		0.92	
28	0.93		0.96	
29	1.00		1.00	
30	1.00		1.00	
31	0.96		1.00	
32	1.00		1.00	
33	0.96		1.00	
34	0.96		0.96	
		0.99		0.98

I-CVI: Item Content Validity Index; S-CVI: Scale Content Validity Index.

**Table 2 healthcare-13-02312-t002:** Description of the sociodemographic characteristics of the sample of 235 first-year students.

	Organic Unit				*p*-Value (Chi-Square Test)
	FAUP	FDUP	FMUP	FPCEUP	
	*n* (%)	*n* (%)	*n* (%)	*n* (%)	
Sex					
Female	35 (74.5)	73 (71.6)	51 (76.1)	17 (89.5)	0.423
Male	12 (25.5)	29 (28.4)	16 (23.9)	2 (10.5)	
Country of High School					
Portugal	38 (80.9)	88 (87.1)	60 (89.6)	15 (78.9)	0.456
Abroad	9 (19.1)	13 (12.9)	7 (10.4)	4 (21.1)	
Type of High School					
Public	30 (63.8)	70 (69.3)	42 (63.6)	11 (61.1)	0.814
Private	17 (36.2)	31 (30.7)	24 (36.4)	7 (38.9)	

FAUP: Faculty of Architecture of the University of Porto; FDUP: Faculty of Law of the University of Porto; FMUP: Faculty of Medicine of the University of Porto; FPCEUP: Faculty of Psychology and Education Sciences of the University of Porto. One participant was excluded from the “Country of High School” analysis because they marked multiple options. Three participants were excluded from the “Type of High School” analysis because they marked multiple answer options.

**Table 3 healthcare-13-02312-t003:** Expectations regarding AI in healthcare (*n* = 235).

	Overall	Sex		*p*-Value (Mann–Whitney Test)	Area of Study		*p*-Value (Mann–Whitney Test)
		Female	Male		Health	Other	
18. AI will improve healthcare.	4 (3; 4)	4 (3; 4)	4 (3; 4)	0.133	4 (3; 4)	4 (3; 4)	0.176
21. I trust AI more than a doctor’s opinion.	1 (1; 2)	1 (1; 2)	1 (1; 2)	0.196	1 (1; 2)	1 (1; 2)	0.566
22. AI will reduce medical errors.	3 (3; 4)	3 (3; 4)	3 (3; 4)	0.951	3 (3; 4)	3 (3; 4)	0.312
23. AI will enable more accurate diagnoses.	4 (3; 4)	4 (3; 4)	4 (3; 4)	0.250	4 (3; 4)	4 (3; 4)	0.579
33. The doctor using AI is responsible for any errors arising from its use.	4 (3; 4)	4 (3; 4)	4 (3; 5)	0.328	4 (3; 4)	4 (3; 5)	0.985
34. AI programmers are responsible for any errors arising from its use.	3 (3; 4)	3 (3; 4)	3 (3; 4)	0.952	3 (3; 4)	3 (3; 4)	0.061

The median, the first, and the third quartiles are shown for each question.

## Data Availability

The datasets generated during and/or analyzed during the current study are available from the corresponding author on reasonable request due to ethical considerations, including protection of participants’ privacy, and prevention of potential misuse outside the proper scientific or ethical context.

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
