# Peer review of "Artificial Intelligence in Healthcare: University Students’ Perceptions and Level of Confidence"

_healthcare, 2025, doi:10.3390/healthcare13182312_

Round 1

Reviewer 1 Report

Comments and Suggestions for Authors

Journal: Healthcare (ISSN 2227-9032)

Manuscript ID healthcare-3782471

Type: Article

Title: Artificial Intelligence in Healthcare: University Students' Perceptions and Level of Confidence.

Authors: Paulo Simões Peres * , Luísa Castro , Ivone Duarte

A brief summary

The topic of this article is both timely and compelling. The authors address a significant issue, particularly as it pertains to a specific population that has been insufficiently studied to date, which constitutes a primary strength of the research. Nonetheless, some minor revisions are necessary. My comments are outlined below:

Line 19 – 21: The information about the time period when the research was conducted is missing; please add it.

Line 28-29: The keywords are not provided according to MeSH criteria, please correct this.

Line 76: Although the introduction is relatively well-written, it should place greater emphasis on what distinguishes this research from previous studies and highlight its significance (for example, that it was conducted on a specific population).

Lines 81-82: Please clarify during which time period the research was conducted.

Line 107: Please clarify during which time period the pilot research (a pre-test) was conducted, as well as the number of participants or the sample size.

Line 133: Please clarify why a power calculation was not conducted to determine the appropriate sample size.

Line 149-151: Please clarify the reasons why consent was not obtained from all the Organic Units (OUs) planned for inclusion in the study.

Line 160-166 and Line 170-172: This part of the text belongs in the section on research limitations, not in the methodology section; please correct it.

Line 173: Please provide a more detailed explanation of which faculties were ultimately included in the study, as well as the inclusion and exclusion criteria for the participants (did the participants sign an informed consent form to take part in the research?).

Line 204: The authors should include a brief statement regarding the response rate (i.e., the percentage of invited participants who agreed to participate) to enable readers to assess the representativeness of the sample.

Lines 219: The distribution of male and female participants is not well balanced. Please clarify which type of statistical tests were applied to account for this variability in the data interpretation.

Line 362: The limitations and shortcomings of the study should be discussed in more detail (e.g. predominantly female participants, study design, limited numbers of included OUs).

Lines 417-481: The reference formatting is inconsistent. All references should be carefully reviewed to ensure full compliance with the journal’s author guidelines. Additionally, it would be beneficial to include more relevant and recent scientific references, preferably published within the last five years (introduction and/or discussion sections).

The manuscript would benefit from further language refinement. Utilizing a professional English-language editing service is recommended.

Comments on the Quality of English Language

The manuscript would benefit from further language refinement. Utilizing a professional English-language editing service is recommended.

Author Response

Thank you for the revision and for raising these concerns. Regarding the comments made:

Comments 1, 4, 5Line 19 – 21: The information about the time period when the research was conducted is missing; please add it; Lines 81-82: Please clarify during which time period the research was conducted; Line 107: Please clarify during which time period the pilot research (a pre-test) was conducted, as well as the number of participants or the sample size.

Response 1, 4, 5: Added to lines 20-21, 115, 180. The number of participants of the pre-test was already mentioned (Line 114)

Comment 2Line 28-29: The keywords are not provided according to MeSH criteria. Please correct this

Response 2: Corrected (Lines 29 and 30)

Comment 3Line 76: Although the introduction is relatively well-written, it should place greater emphasis on what distinguishes this research from previous studies and highlight its significance (for example, that it was conducted on a specific population).

Response 3: Rewritten (Lines 77 to 81)

Comment 6Line 133: Please clarify why a power calculation was not conducted to determine the appropriate sample size.

Response 6: No power calculation was performed to determine the sample size, as this was a pilot study, intended to provide preliminary insights into students’ perceptions of AI in healthcare and inform future research with larger and more representative samples (lines 163-168). The margin of error was also calcuted considering the concerns raised.

Comment 7Line 149-151: Please clarify the reasons why consent was not obtained from all the Organic Units (OUs) planned for inclusion in the study.

Response 7: Mentioned in lines 153 and 160. The reason to not delivering the study in said OUs was due to the lack of response to the contact made by the researchers or the response arrived too late.

Comment 8Line 160-166 and Line 170-172: This part of the text belongs in the section on research limitations, not in the methodology section; please correct it.

Response 8: Corrected, mentioned in the Limitations section.

Comment 9Line 173: Please provide a more detailed explanation of which faculties were ultimately included in the study, as well as the inclusion and exclusion criteria for the participants (did the participants sign an informed consent form to take part in the research?).

Response 9: Further clarified (Lines 182 to 186)

Comment 10Line 204: The authors should include a brief statement regarding the response rate (i.e., the percentage of invited participants who agreed to participate) to enable readers to assess the representativeness of the sample.

Response 10: Added in the Results section (Lines 215 and 216)

Comment 11Lines 219: The distribution of male and female participants is not well balanced. Please clarify which type of statistical tests were applied to account for this variability in the data interpretation.

Response 11: The sample was not gender-balanced and non-parametric tests (Mann-Whitney U and Kruskal-Wallis) were used for group comparisons. These tests were employed because of the ordinal nature of the items, and are also appropriate for unequal group sizes, helping to minimize bias in the analysis.

Comment 12Line 362: The limitations and shortcomings of the study should be discussed in more detail (e.g., predominantly female participants, study design, limited numbers of included OUs).

Response 12: Limitations section rewritten

Comment 13Lines 417-481: The reference formatting is inconsistent. All references should be carefully reviewed to ensure full compliance with the journal’s author guidelines. Additionally, it would be beneficial to include more relevant and recent scientific references, preferably published within the last five years (introduction and/or discussion sections).

Response 13: The formatting of the references was made using EndNote and the template provided by MDPI/Healthcare. Therefore, we could not find the references formatted incorrectly. More recent articles were added as references, reflecting the current pertinence of the topic.

Reviewer 2 Report

Comments and Suggestions for Authors

The article addresses a timely and relevant topic: university students' perceptions of AI in healthcare. While the manuscript presents a structured survey and reflects preliminary insights, there are several important areas that need clarification, expansion, or refinement before it can be considered for publication.

The topic addressed in the article is both contemporary and increasingly important within the context of healthcare education and ethics. The authors have ensured ethical rigor by obtaining proper approval and validating the questionnaire used, which adds credibility to the research process. Furthermore, the article presents a sufficiently structured introductory framework and contextualizes the results by drawing comparisons with existing literature, thereby enhancing its relevance and grounding it within the broader academic discourse.

Literature Review and Referencing:
The literature cited is limited in scope, particularly in recent, high-impact studies from 2023–2025 on AI in health education.
Consider expanding the background with a more detailed review of similar studies on student perceptions internationally.
Some references are repeated across sections or too general (e.g., WHO digital strategy) without deeply informing the research context.
Suggested improvement: Increase the number of recent and specialized references, especially those concerning AI literacy in higher education, comparative cross-cultural studies, and AI ethics in medical training.

Methodological Detail:
The methodology is described, but important aspects are missing or underdeveloped:
The questionnaire itself is only briefly summarized. Key items should be provided within the article or detailed in supplementary materials.
The sampling method is convenience-based, which is acceptable for exploratory studies, but this limitation should be emphasized more explicitly.
There is no explanation of response rate or how many students were invited vs. participated.
The statistical analysis section is appropriate, though overly technical in places. A summary of how statistical significance was interpreted should be added for accessibility.

Results Presentation:
The results are generally well-structured, but figures and tables are very limited.
Table 3 includes multiple Likert-based variables but lacks clear visualization (e.g., bar charts or graphs) to support interpretation.
Important findings such as gender or faculty-based differences are discussed, but non-significant results dominate the discussion, limiting the article’s contribution.
Suggested improvement: Include figures (bar charts, stacked graphs, etc.) to better illustrate key variables like familiarity with AI, trust levels, and perceived responsibilities.

Discussion and Contribution:
While the discussion links results to literature, it often repeats rather than interprets findings.
The novelty or unique contribution of this study is not clearly stated — especially since similar studies have been conducted in Portugal and elsewhere.
The ethical implications of AI usage (e.g., automation bias, black box systems) are introduced briefly, but lack depth.
Suggested improvement: Enhance discussion with critical reflection on implications for curriculum development and AI policy in higher education.

Conclusion:
The conclusion effectively summarizes the main findings but is somewhat general.
The educational implications of the findings should be more prominently emphasized.
Consider including a call to action or concrete proposals for improving AI literacy among university students.

Formal and Structural Issues:
The flow of some sections (particularly methods and discussion) can be improved for clarity.
Minor English language issues and some awkward phrasings are present throughout the manuscript. A professional English language editing would enhance readability.
There are repetitions (e.g., of statistical results in both results and discussion) that could be streamlined.

Comments on the Quality of English Language

Minor English language issues and some awkward phrasings are present throughout the manuscript. A professional English language editing would enhance readability.

Author Response

Thank you for the revision and raising these concerns. Regarding the comments made:

Literature Review and Referencing:
The literature cited is limited in scope, particularly in recent, high-impact studies from 2023–2025 on AI in health education.
Consider expanding the background with a more detailed review of similar studies on student perceptions internationally.
Some references are repeated across sections or are too general (e.g., WHO digital strategy) without deeply informing the research context.
Suggested improvement: Increase the number of recent and specialized references, especially those concerning AI literacy in higher education, comparative cross-cultural studies, and AI ethics in medical training.

Response: We added more recent references, highlighting the scope of the existing evidence on the topic (Reference 15, 26 and 29) (Line 67, 337, 359)

Methodological Detail:
The methodology is described, but important aspects are missing or underdeveloped:
The questionnaire itself is only briefly summarized. Key items should be provided within the article or detailed in supplementary materials.
The sampling method is convenience-based, which is acceptable for exploratory studies, but this limitation should be emphasized more explicitly.
There is no explanation of response rate or how many students were invited vs. participated.
The statistical analysis section is appropriate, though overly technical in places. A summary of how statistical significance was interpreted should be added for accessibility.

Response: The full questionnaire is provided in supplementary materials. The sampling process is mentioned as the main limitation and further explaind in the Limitations section. As suggested, we added the response rate in the Results section. To improve accessibility by a more diverse readership (including clinicians and health researchers), we added a summary at the end of the Statistical Analysis section, clarifying how statistical significance was interpreted

Results Presentation:
The results are generally well-structured, but figures and tables are very limited.
Table 3 includes multiple Likert-based variables but lacks clear visualization (e.g., bar charts or graphs) to support interpretation.
Important findings such as gender or faculty-based differences are discussed, but non-significant results dominate the discussion, limiting the article’s contribution.
Suggested improvement: Include figures (bar charts, stacked graphs, etc.) to better illustrate key variables like familiarity with AI, trust levels, and perceived responsibilities.

Response: As suggested, we added graphs to clearly illustrate the distribution of answers of participants by sex and field of study.

Discussion and Contribution:
While the discussion links results to literature, it often repeats rather than interprets findings.
The novelty or unique contribution of this study is not clearly stated, especially since similar studies have been conducted in Portugal and elsewhere.
The ethical implications of AI usage (e.g., automation bias, black box systems) are introduced briefly, but lack depth.
Suggested improvement: Enhance discussion with critical reflection on implications for curriculum development and AI policy in higher education.

Response: As suggested, we further clarified the implications for the curricula in several sections: Discussion (Literacy and knowledge – lines 310, 311, 31-315 –, and Liability and ethical – Lines 359-362) and Conclusion

Conclusion:
The conclusion effectively summarizes the main findings but is somewhat general.
The educational implications of the findings should be more prominently emphasized.
Consider including a call to action or concrete proposals for improving AI literacy among university students.

Response: As suggested, we included suggestions for future studies and possible implications in the AI field. (Lines 419 – 428)

Formal and Structural Issues:
The flow of some sections (particularly methods and discussion) can be improved for clarity.
Minor English language issues and some awkward phrasings are present throughout the manuscript. A professional English language editing would enhance readability.
There are repetitions (e.g., of statistical results in both results and discussion) that could be streamlined.

Response: We thank the reviewer for these constructive comments. Following the suggestions, we have revised the flow of the Methods and Discussion sections to improve clarity and readability; Minor English language issues and awkward phrasing have been corrected throughout the manuscript. A thorough language editing has been performed to enhance fluency; Repetitions, particularly the statistical results presented both in the Results and Discussion sections, have been streamlined to avoid redundancy.

We believe these changes have improved the overall clarity and readability of the manuscript.

Round 2

Reviewer 2 Report

Comments and Suggestions for Authors

The authors have responded thoroughly to the points raised in the first round of review, and I think the manuscript has improved in clarity, depth, and readability. 
The manuscript will be substantially improved and suitable for publication after minor additional refinement. Specifically, I recommend ensuring that the most recent high-impact systematic reviews are included in the literature review, and slightly condensing the discussion where it still repeats results.

Author Response

We thank the reviewer for the comments made, that were taken into account. 

Regarding the inclusion of "high-impact systematic reviews", we included one in the Introduction (reference 16) and another in the Discussion (reference 25).

We also reduced the repetition of results in the discussion, namely in section 4.1 and 4.2.

We hope these updates were able to answer your concerns.